# Comprehensive Recovery Technology for Te, Au, and Ag from a Telluride-Type Refractory Gold Mine

**Wei Yang [1,2,*], Gang Wang [1,2,*], Qian Wang [1,2], Ping Dong [1], Huan Cao [1] and Kai Zhang [1]**

1   School of Resources Engineering, Xi'an University of Architecture and Technology, Xi'an 710055, China; wangqianooo@outlook.com (Q.W.); dongping0318@163.com (P.D.); Hanniecao@126.com (H.C.); kaizhangjack@126.com (K.Z.)

2   Key Laboratory of Gold and Resources in Shaanxi Province, Xi'an 710055, China

*   Correspondence: yangwei@xauat.edu.cn (W.Y.); wghbdxk@126.com (G.W.)

**Abstract:** While extracting gold and silver from telluride-type gold deposits, it is beneficial to develop a comprehensive recovery technology for tellurium. In this paper, we report process mineralogy based on the backward processing technology and the low comprehensive utilization rate of typical telluride-type gold deposits in Xiaoqinling, China. The findings show that tellurium, gold, and silver are the most valuable elements in the ore fissures and gangue minerals and are encapsulated in metallic sulfur ore in the form of altaite, hessite, calaverite, antamokite and natural gold. The flotation method was innovatively applied in this study to comprehensively recover Te, Au and Ag. The results show that when the ore particle size was −0.074 mm (70%), the flotation pulp density was 33%, the pulp pH was 8, and the combined collector (isoamyl xanthate + ethyl thio- carbamate (1:1)) was 120 g/t, in the process involving one rough flotation step, two cleaning flotations and two scavenging flotations as well as a continuous 8 d industrial test, the recovery degree was stable and the average grades of Te, Au, and Ag were 241.61, 90.30, and 92.74 g/t with 95.42%, 97.28%, and 94.65% recovery rates, respectively; thus, excellent recovery degrees were obtained. Compared with the original flotation process, the recovery rates of Te, Au, and Ag were increased by 19.91%, 6.93%, and 5.67%, which boosted the effective enrichment of all valuable elements in the telluride-type gold mine and achieved technological progress.

**Keywords:** telluride-type gold mine; flotation; tellurium; comprehensive recovery

## 1. Introduction

Rare elements have received increasing attention in recent years as they play important roles in modern high-tech industries and national defense [1–4]. However, the average abundance of tellurium in the crust is only 0.005 ppm [5], and there are no independent deposits of it. At present, tellurium is mainly extracted from tellurium-rich copper and lead smelting slag [6–9]. These processes are not only complex and time-consuming, but also result in great losses of tellurium during the recovery process.

There are a large number of telluride-type gold deposits in Xiaoqinling, China, which content of tellurium is relatively high [10,11]. To recover this kind of ore, the direct recovery of tellurium from ore by flotation can both reduce the loss of tellurium and improve the utilization rate of ore, which is of great significance. Tellurium-bearing gold deposits are an important type of gold ore. Among the 19 known types of industrial gold minerals, tellurium-bearing gold minerals account for 11 of them [12]. So far, gold and silver are mainly recovered from telluride-type gold deposits, but there has been no report of tellurium recovery from such sources. Even more problematic is that, for telluride-type gold deposits, the existing flotation system for recovering Au and Ag mostly applies xanthate as the collector and CaO as the regulator, which can only recover tellurium closely related to Au and Ag.

The flotation process of tellurobismuthite is similar to that of pyrites [13]. Inhibitors are often used to inhibit pyrite [14], so as to promote the recovery of tellurobismuthite. However, such a reagent system can result in the loss of gold and silver closely related to pyrite. Moreover, the brittleness and sliming of telluride [15–17] will further increase the difficulty of Te recovery. Therefore, it is of great value to develop a flotation process for recovering Te from telluride-type gold ores, while taking into account the recovery of Au and Ag.

Aiming at decreasing the difficulty of comprehensively utilizing a telluride-type gold deposit in the Xiaoqinling, China, single factor condition test as well as open-circuit and closed-circuit tests were conducted by the flotation method on the basis of process mineralogy. Once the optimum process conditions were set, industrial tests of 300 t/d throughputs were completed, and excellent efficiency was obtained, which, in turn, led to a comprehensive recovery of Te, Au and Ag from telluride-type gold deposits. This research provides a feasible method of extracting valuable elements from telluride-type gold deposits.

## 2. Experimental Section

### 2.1. Experimental Sample and Experimental Reagents

Telluride-type gold mine samples used in the flotation tests were obtained from Xiaoqinling, China. In the test, raw samples, after they were crushed to less than 3 mm in a roller crusher, were finely ground to −0.074 mm (60–80%) with a ball mill before flotation. Butyl xanthate (industrial grade), sodium isoamyl xanthate (industrial grade), ammonium dibutyl dithiophosphate (industrial grade) and ethyl thiocarbamate (industrial grade) were used as collectors. Butyl xanthine and sodium isoamyl xanthate are commonly used as sulfide ore collectors; ammonium dibutyl dithiophosphate can improve the recovery of refractory ores, and ethyl thiocarbamate is often used in the flotation of galena. Sodium hydroxide (analytical grade) was used as the pH modifier, and terpineol (industrial grade) was used as the frother.

### 2.2. Experimental Methods

Bench-scale flotation experiments were performed in a 1.5-L XFD (Nanchang Haifeng Mining Equipment Co, Ltd., Nanchang, China) flotation cell. Ore sample (500 g) was added to an XMQ (Nanchang Haifeng Mining Equipment Co, Ltd., Nanchang, China) ball mill with a pulp density of 60 wt. %, and the samples were placed in a flotation cell after they were ground, and the amount of solid minerals was 33 wt. %. When the flotation machine was open-stirred for 3 min while pH was adjusted, a collector and the frother were added to the pulp. The dosage of the collector and the foaming reagent in the scavenging flotation was half of the dosage of the roughing reagent. A diagram of the single factor test is shown in Figure 1.

After the test, the collected products were filtered, dried and weighed; the concentrations of Te, Au, and Ag in the samples were determined, and the recovery rate was calculated. The recovery rate amount was then calculated by using the expression:

$$\varepsilon = \beta\gamma/\beta_1 \tag{1}$$

where $\varepsilon$ is the degree of recovery for a specific element (%); $\beta$ is the grade of the element in the product (g/t); $\gamma$ is the yield of the product (%) and $\beta_1$ is the grade of feed minerals (g/t).

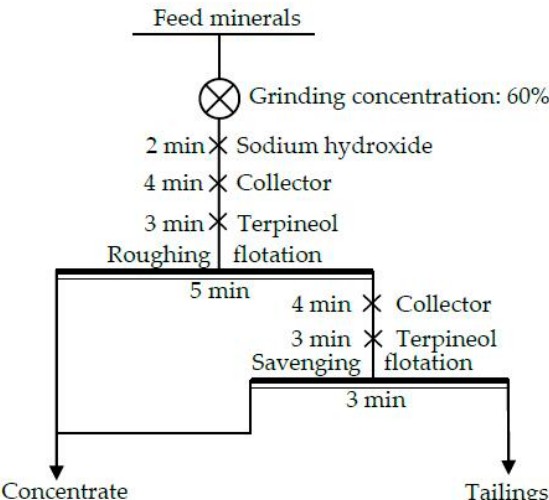

**Figure 1.** Chart of the single factor condition test.

*2.3. Analytical Method*

With the help of an OPTON mineral liberation analyzer (MLA, Beijing Opton Optical Technology Co., Ltd., Beijing, China) and an APOLLO X-ray energy dispersive spectrometer (Denver, CO, USA), the mineral composition, particle size and ore association relation of the telluride-type gold mine experimental sample were determined, and the MLA operating voltage is 25 kV during the test. With the help of an S4 Pioneer wavelength dispersive X-ray Fluorescence (XRF) spectrometer (Bruker, Germany), the mineral elements and content of the mine experimental sample were determined, and the test sample needs to be in a dry state, the weight is above 3 g, and the fineness is –0.074 mm. With the help of a MM-7C polarizing microscope (Shanghai Wanheng Precision Instrument Co, Ltd., Shanghai, China), the ore association relation of the mine experimental sample was determined. With the help of a PerkinElmer NexION 300X inductively coupled plasma mass spectrometry (ICP-MS) ((Boston, MA, USA), the content of elements of the mine experimental sample were determined, and the test process RF power is 1300 W and the auxiliary gas is 0.7 L/min. With the help of an Ultima IV X-ray diffraction X-ray diffractometer (XRD) (Rigaku Corporation, Tokyo, Japan), fitted with a copper tube (copper K$\alpha$ radiation). The mineral compositions of the mine experimental sample were determined, and the data were analyzed with software (jade 6.5) and the mineral content was calculated. The test sample weight is above 3 g, and the fineness is –0.074 mm, the test process is set to 5–90° and the scanning rate is 5°/min.

## 3. Results and Discussion

*3.1. Mineralogical Analysis*

With the help of a MLA, the telluride-type gold mine experimental sample was analyzed, and the results are shown in Table 1. The results of observation and analysis under a polarizing microscope are shown in Figure 2. With the help of XRF and ICP-MS, the chemical multi-element analysis of the telluride-type gold mine experimental sample was determined and the results are shown in Table 2.

A mineralogical study showed the following: (1) there were 11 ore minerals in the sample, five of which are Au, Ag, and Te minerals, namely natural gold, antamokite, calaverite, hessite, and altaite, which are major target minerals; (2) the other metal sulfide minerals were pyrite, chalcopyrite, bornite, galena, blende, and pyrrhotite; (3) Au, Ag, and Te mainly exist in ore fissures and gangue minerals and are encapsulated in pyrite and other metal sulfide minerals; (4) hessite and altaite are closely related to galena and natural gold is closely related to quartz. This occurrence state is beneficial for the recovery of Au, Ag, and Te. The particle size of Au, Ag, and Te minerals is small, all below 0.090 mm. The particles of antamokite and altaite are relatively coarse, while the particle sizes of natural gold, calaverite, and

hessite all have particle sizes below 0.038 mm. The particle size of pyrite, galena and chalcopyrite is larger than 0.016 mm, and some of them have reached 1–2 mm, which is better sufficient for grinding dissociation. The gold ore is primary deposit and contains polymetallic sulfide with a quartz vein, which is easily recovered by flotation technology. Simultaneously, the grades of tellurium, gold, and silver are 14.80, 5.30, and 5.70 g/t respectively, according to multi-element analysis. These are the main recovered elements. The concentrations of the other elements are too low to have a recovery value.

**Table 1.** Mineral composition, chemical formula, and dissemination size of the raw ore.

| Mineral Composition | Chemical Formula | Dissemination Size (mm) |
|---|---|---|
| Natural gold | Au | <0.038 |
| Calaverite | $AuTe_2$ | <0.038 |
| Hessite | $Ag_2Te$ | <0.038 |
| Altaite | PbTe | 0.008–0.09 |
| Antamokite | $Ag_3AuTe_2$ | 0.006–0.09 |
| Pyrite | $FeS_2$ | 0.048–2 |
| Galena | PbS | 0.037–2 |
| Chalcopyrite | $CuFeS_2$ | 0.016–1 |
| Bornite | $Cu_5FeS_4$ | - |
| Sphalerite | ZnS | - |
| Pyrrhotite | $Fe_{1-x}S$ | - |
| Quartz | $SiO_2$ | >0.005 |

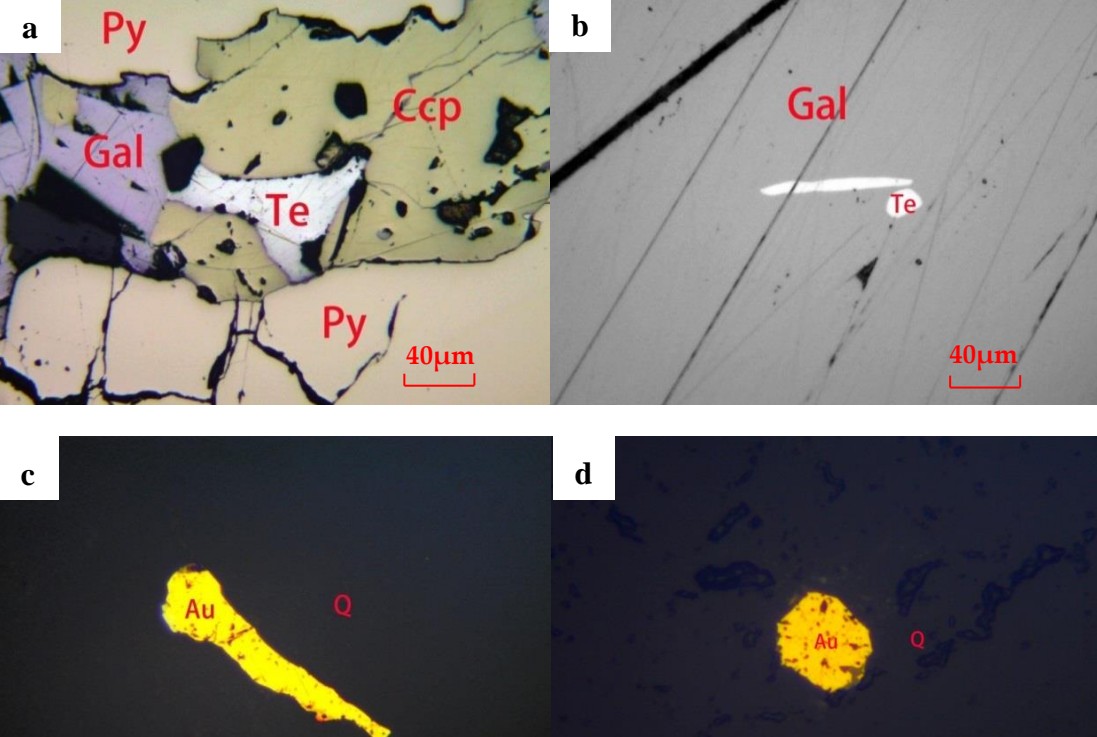

**Figure 2.** Ore association: (**a**) Long strips and round grains of fine-grained altaite and galena co-existing and interspersed with chalcopyrit; (**b**) long strips and round granular fine-grained altaite encased in galena; (**c**) a long strip of natural gold enclosed in late quartz; (**d**) circular grain of natural gold enclosed in late quartz.1cm (Py—pyrite; Gal—galena; Te—altaite; Ccp—chalcopyrite; Au—natural gold; Q—quartz).

**Table 2.** Percentage concentrations of multi-element minerals.

| Component | SiO$_2$ | TiO$_2$ | Al$_2$O$_3$ | TFe | Mn | MgO | CaO | K$_2$O |
|---|---|---|---|---|---|---|---|---|
| **Content (%)** | 74.92 | 0.28 | 6.38 | 2.35 | 0.076 | 1.06 | 1.71 | 3.76 |
| **Component** | Na$_2$O | Cu | Pb | Zn | S | Au* | Ag* | Te * |
| **Content (%)** | 0.84 | 0.013 | 0.1 | 0.03 | 2.76 | 5.30 | 5.70 | 14.80 |

"*" means that the unit is g/t.

### 3.2. Study on Flotation Efficiency

#### 3.2.1. Influence of Ore Particle Size on the Recovery of Te, Au, and Ag

Figure 3 shows the results of the effect of ore particle size on the recovery of Te, Au, and Ag. As shown in Figure 3, with the increase in ore particle size, the recovery of Au, Ag, and Te first increases and then decreases. The recovery of Au, Ag, and Te reaches 92.35, 94.32, and 92.09%, respectively, at a particle size of −0.074 mm (70%). A continuous increase in ore particle size will result in a significant decrease in the recovery of all minerals, and this is due to the fact that telluride is liable to be over-ground, resulting in slimming and making it difficult for flotation to occur. Therefore, the optimum ore particle size is −0.074 mm (70%).

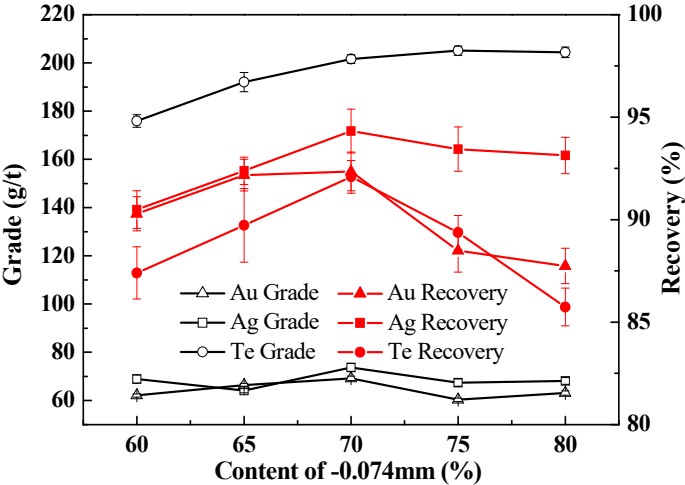

**Figure 3.** Flotation recovery and grade of Te, Au, and Ag as a function of the ore particle size value (pulp density: 33%; pulp pH value: 8; butyl xanthate: 120 g/t).

#### 3.2.2. Effect of Pulp pH on the Recovery of Te, Au, and Ag

The effect of pulp pH on the recovery of Te, Au, and Ag shown in Figure 4. As shown in Figure 4, pH has the highest effect on the recovery of Au, but has little effect on the recovery of Ag. The recovery rate of Au first increases and then decreases with the rise in pH value, reaching a maximum of 93.91% when pH = 8.

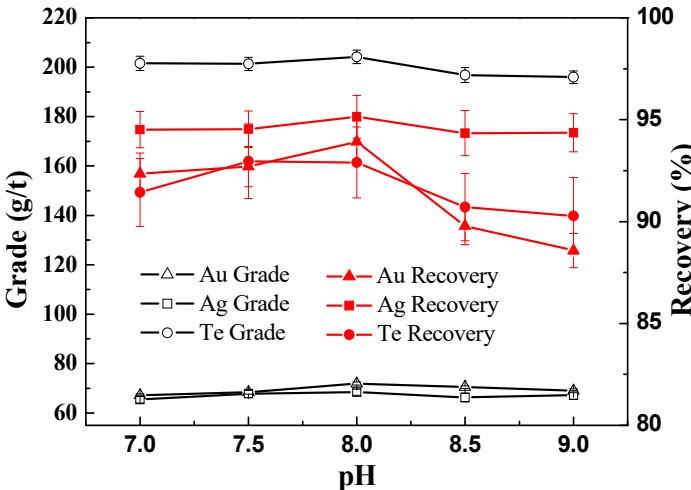

**Figure 4.** Flotation recovery and grade of Te, Au, and Ag as a function of pulp pH value (content of −0.074 mm (70%); pulp density: 33%; butyl xanthate: 120 g/t).

Moreover, 95.14% Ag is obtained at this point, and 92.96% Te is recovered when pH = 7.5. Since Au and Ag are the most valuable elements, and since the recovery rate of Te at pH 8 is similar to that at pH 7.5, a pH value of 8 was selected for the slurry as the optimal pH for subsequent experiments.

### 3.2.3. Effect of Collector Types on the Recovery of Te, Au, and Ag

The effect of collector type on the recovery of Te, Au and Ag is shown in Table 3. The results in the table are the mean values after three tests. The products are concentrates. It is shown in Table 3 that, when ammonium dibutyl dithiophosphate was used as a collector, the concentrate yield was the highest, reaching 14.18%; the grades of Te, Au, and Ag were 94.73, 29.62, and 37.88 g/t, but the recovery rates were only 90.88, 80.32, and 94.57%. The low grade and recovery rates indicate that ammonium dibutyl dithiophosphate has a strong collecting performance but poor selectivity, and it is not suitable for telluride-type gold ore. Compared with butyl xanthate, sodium isoamyl xanthate has a better flotation effect. Te, Au, and Ag were relatively low in grade, but the recovery rate was obviously high. This is because the collection ability and selectivity of xanthate are closely related to the length of the hydrocarbon chain in its molecule [18–20]. When the number of carbon atoms is small ($C_1$–$C_6$), the longer the carbon chain is, the stronger the collection performance of xanthate will be, but the selectivity of xanthate is the opposite—the longer the carbon chain is, the worse the selectivity will be. Compared with sodium isoamyl xanthate and ethyl thiocarbamate, isoamyl xanthate has a higher recovery of Au, while ethyl thiocarbamate has a higher recovery of Ag and Te, which is because hessite and altaite are closely related to galena. As an effective collector of galena, ethyl thiocarbamate [21] can significantly improve the recovery of Ag and Te, but it is not suitable to recover gold because it is not closely related to galena. When sodium isoamyl xanthate and ethyl thio- carbamate are used as a combined collector, all valuable elements in this type of ore can be recovered. The recovery of Te, Au, and Ag is better than with a single reagent. The grades of Te, Au, and Ag that can be reached are 190.93, 68.34, and 73.45 g/t, and the recovery rates are 95.08%, 95.03%, and 94.98%, respectively.

**Table 3.** Flotation recovery and grade of Te, Au, and Ag of collector type value.

| Collector Type | Yield (%) | Grade (g/t) | | | Recovery (%) | | |
|---|---|---|---|---|---|---|---|
| | | Au | Ag | Te | Au | Ag | Te |
| Butyl xanthate | 6.91 | 72.27 | 76.70 | 195.5 | 93.88 | 93.28 | 91.95 |
| Sodium isoamyl xanthate | 7.27 | 70.28 | 72.03 | 190.31 | 95.33 | 94.64 | 92.73 |
| Ammonium dibutyl dithiophosphate | 14.18 | 29.62 | 37.88 | 94.73 | 80.32 | 94.57 | 90.88 |
| Ethyl thiocarbamate | 7.45 | 53.39 | 70.03 | 186.65 | 71.66 | 95.27 | 95.31 |
| Sodium isoamyl xanthate and ethyl thiocarbamate (1:1) | 7.37 | 68.34 | 73.45 | 190.93 | 95.03 | 94.98 | 95.08 |

(Test conditions: content of −0.074 mm (70%); pulp density: 33%; pulp pH: 8; amount of collector: 120 g/t).

### 3.2.4. Effect of Collector Dosage on the Recovery of Te, Au, and Ag

The effect of collector dosage on the recovery of Te, Au, and Ag is shown in Figure 5. As can be seen, the recovery rate of Te, Au, and Ag first increases and then tends to be stable with the rise of in collector dosage. When the dosage of the collector is 120 g/t, the recovery rates of Te, Au and Ag are 92.43%, 95.43%, and 96.69%. When the dosage of the collector is 150 g/t, the recovery rate decreases slightly, so the suitable dosage of the combined collector (sodium isoamyl xanthate + ethyl thio-carbamate (1:1)) is 120 g/t.

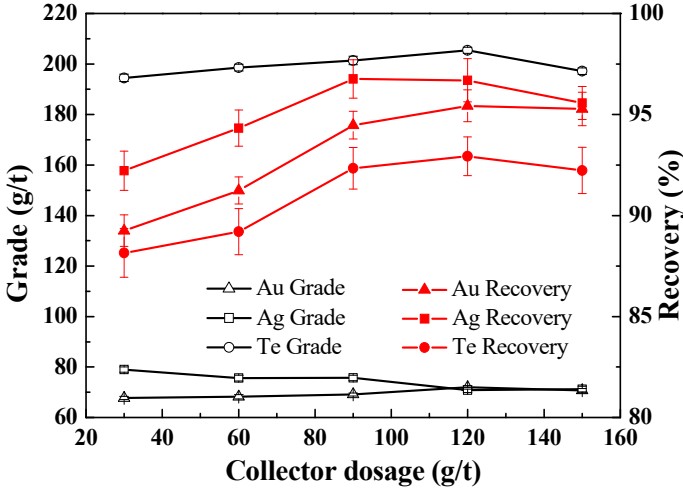

**Figure 5.** Flotation recovery and grade of Te, Au, and Ag as a function of collector dosage value (content of −0.074 mm (70%); pulp density: 33%; pulp pH: 8; combined collector: sodium isoamyl xanthate + ethyl thiocarbamate (1:1)).

### 3.2.5. Effect of Slurry Pulp Density on the Recovery of Te, Au and Ag

The effect of slurry concentration on the recovery of Te, Au, and Ag is shown in Figure 6. It can be seen in Figure 6 that the grades of Te and Ag first increase and then decrease with the increase in pulp density, and reach the highest when the pulp density is 33%; however, the grade of Au gradually increases. When the concentration of the pulp is too low, the probability of collision between bubbles and minerals will be reduced, which is not conducive to the flotation of the target minerals, and the consumption of the reagents is increased; when the concentration of the slurry reaches 33%, the recovery rates of Au, Ag, and Te are 95.84%, 97.10%, and 93.32%, slightly higher than the recovery rate when the density is 35%, so the flotation pulp density was finally set to 33%.

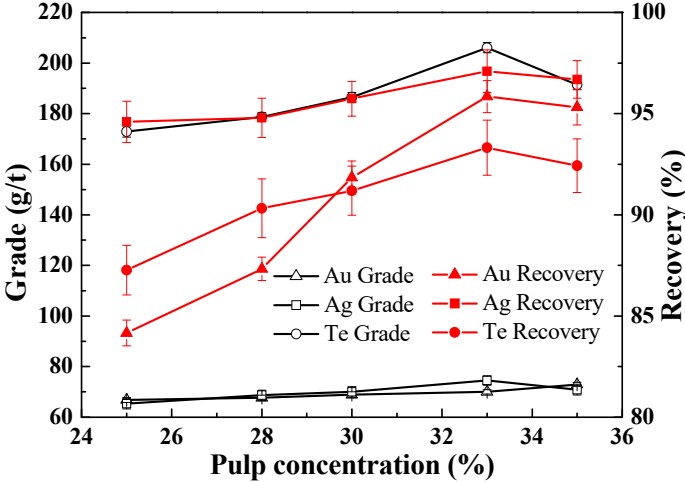

**Figure 6.** Flotation recovery and grade of Te, Au, and Ag as a function of pulp concentration value (content of −0.074 mm (70%); pulp pH: 8; combined collector: sodium isoamyl xanthate + ethyl thio-carbamate (1:1) 120 g/t).

### 3.2.6. The Closed-Circuit Flotation Test

A closed-circuit test was carried out on the basis of the conditions of the above tests. The process chart is shown in Figure 7, and the test results are shown in Table 4. It is shown in Table 4 that, through a closed-circuit flotation test with one rough flotation step, two cleaning flotations and two scavenging flotation processes, Te, Au, and Ag concentrates with grades of 242.95, 89.30, and 93.00 g/t can be obtained, and the recovery rates are 95.54%, 97.18%, and 94.96% respectively. The target minerals were basically recovered.

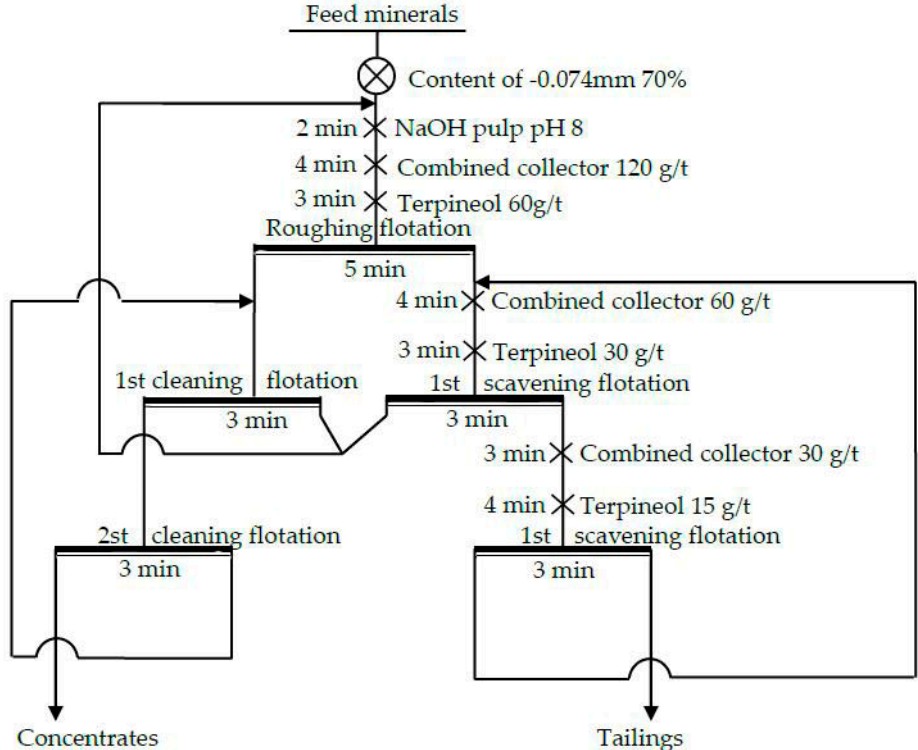

**Figure 7.** Chart of a closed-circuit flotation test.

**Table 4.** Results of a closed-circuit flotation test.

| Product | Yield (%) | Grade (g/t) | | | Recovery (%) | | |
|---|---|---|---|---|---|---|---|
| | | **Au** | **Ag** | **Te** | **Au** | **Ag** | **Te** |
| Concentrates | 5.82 | 89.30 | 93.00 | 242.95 | 97.18 | 94.96 | 95.54 |
| Final tailings | 94.18 | 0.16 | 0.31 | 0.70 | 2.82 | 5.04 | 4.46 |
| Feed minerals | 100.00 | 5.30 | 5.70 | 14.80 | 100.00 | 100.00 | 100.00 |

### 3.3. Industrial Tests

On the basis of the above tests, continuous industrial tests of a 300 t/d concentrator in Xiaoqinling in China were conducted for eight days. Figure 8a presents the actual process operating conditions and production indicators obtained at the production site of a concentrator in Xiaoqinling. Figure 8b presents the industrial test results after the optimization of technological conditions in the study.

In Figure 8a, the average grades of Te, Au, and Ag were 195.61, 83.90, and 88.98g/t, and the recovery rates were 75.44%, 90.35%, and 89.07%. After adopting the new process, in Figure 8b, the average grades of Te, Au, and Ag were 241.61, 90.02, and 92.74 g/t, and the recovery rates were 95.35%, 97.28% and 94.74%, the recovery rates of Te, Au, and Ag increased by 19.91%, 6.93% and 5.67%. Thus, excellent flotation efficiency was obtained, and comprehensive recovery of Te, Au, and Ag in a telluride-type gold mine was achieved.

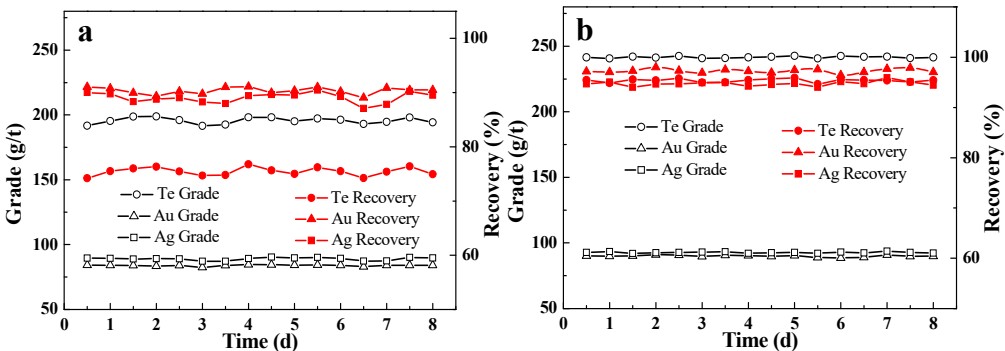

**Figure 8.** (**a**) Production data obtained from the operation site of a 300 t/d concentrator in Xiaoqinling: content of −0.074 mm (66%); CaO adjusts pulp pH: 9; collector: sodium isoamyl xanthate 90 g/t; roughing pulp density: 33%; (**b**)The 300 t/d Industrial test results after optimization of process conditions: content of −0.074 mm (70%); NaOH adjusts pulp pH: 8; combined collector: sodium isoamyl xanthate + ethyl thiocarbamate (1:1) 120 g/t; pulp density: 33%.

### 3.4. Flotation Product Analysis

The flotation concentrates and tailings crystal phase composition in the industrial tests was determined by XRD, whose detailed processes and methods have been previously described by Han [22]. The data were analyzed with software (jade 6.5, Modern Dispersions, Convent, LA, USA) to calculate the content of minerals in the ore. The results of concentrate analysis are shown in Figure 9, Table 5 and tailings analysis is shown in Figure 10, Table 6.

It is shown in Figure 9 and Table 5 that pyrite is the main mineral in flotation concentrate, with a high content of 84.34%. It also contains a small amount of galena and pyrrhotite, but the gangue mineral content is small. As shown in Figure 10 and Table 6, gangue minerals such as quartz, muscovite and potassium feldspar are the main types of flotation tailings. Comparing the analysis results of flotation concentrate and flotation tailings, it can be inferred that the minerals that can be comprehensively recovered and utilized, as well as the main objective minerals, have mostly been enriched in flotation concentrate, and the flotation effect is satisfactory.

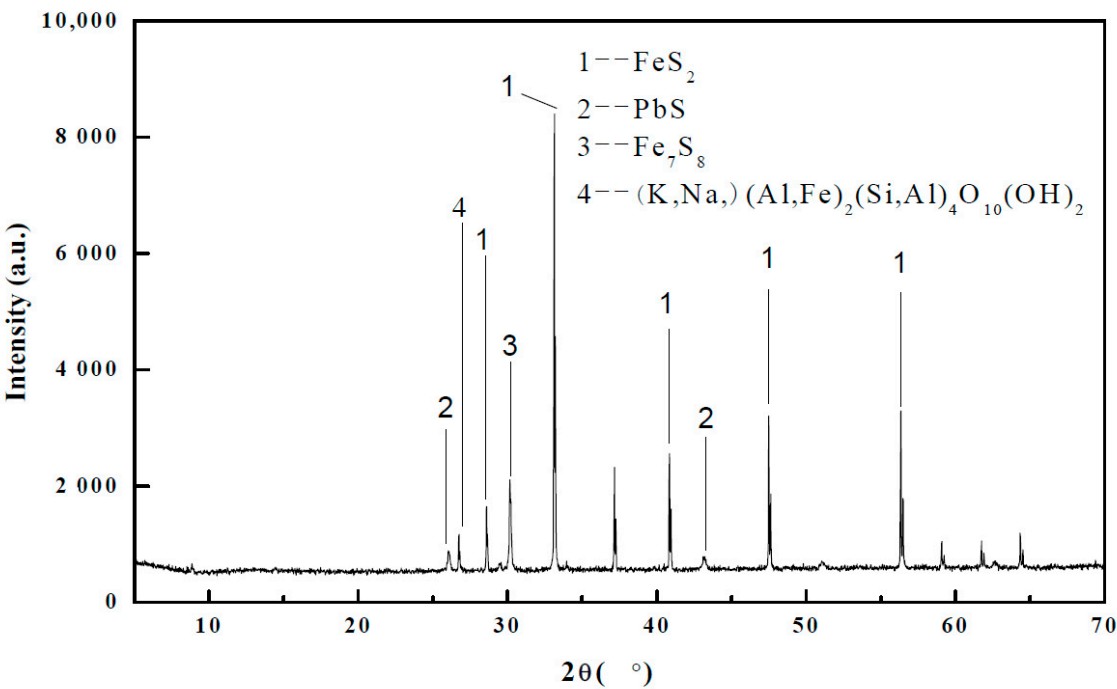

**Figure 9.** XRD detection and analysis of concentrate composition.

**Table 5.** Mineral composition content of concentrate.

| Mineral Composition | Chemical Formula | Content (%) |
| --- | --- | --- |
| Pyrite | $FeS_2$ | 84.34 |
| Galena | PbS | 1.20 |
| Pyrrhotite | $Fe_{1-x}S$ | 1.45 |
| Muscovite | $(K,Na)(Al,Fe)_2(Si,Al)_4O_{10}(OH)_2$ | 13.01 |

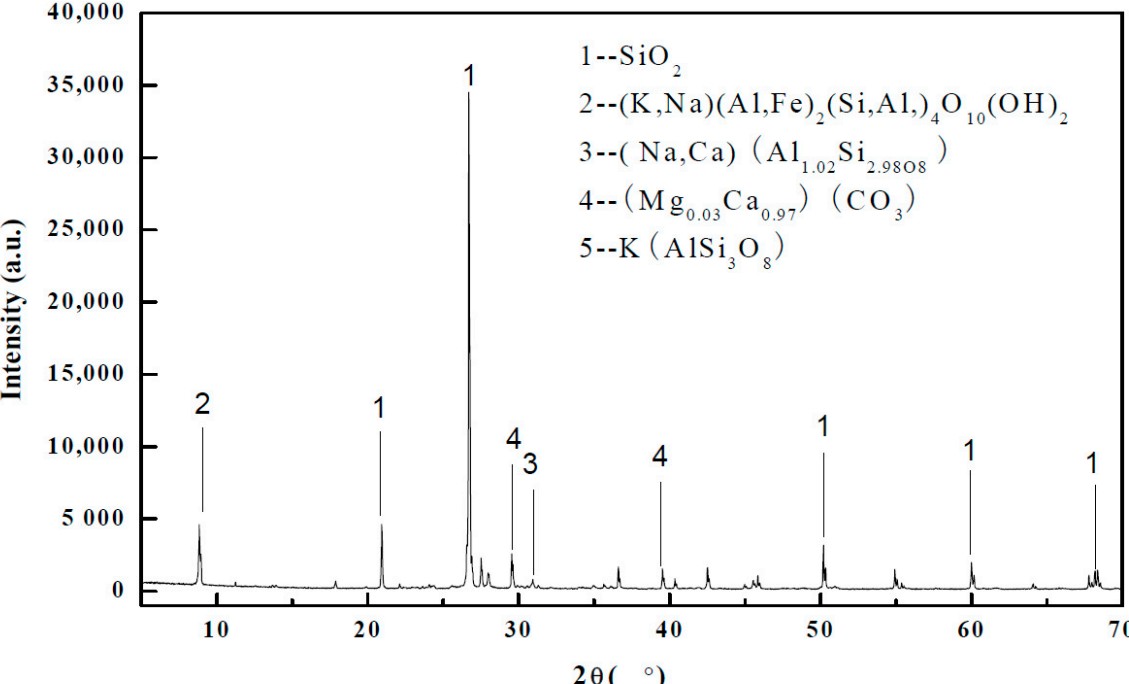

**Figure 10.** XRD detection and analysis of the tailings composition.

**Table 6.** Mineral composition content of tailings.

| Mineral Composition | Chemical Formula | Content (%) |
|---|---|---|
| Quartz | $SiO_2$ | 24.87 |
| Muscovite | $(K,Na)(Al,Fe)_2(Si,Al)_4O_{10}(OH)_2$ | 30.94 |
| Plagioclase | $(Na,Ca)(Al_{1.02}Si_{2.98}O_8)$ | 10.82 |
| Calcite | $(Mg_{0.03}Ca_{0.97})(CO_3)$ | 6.38 |
| Potash feldspar | $KAl_2(Si_3Al)O_{10}(OH,F)_2$ | 26.98 |

## 4. Conclusions

There were five kinds of Au, Ag, and Te minerals in the original ore: natural gold, antamokite, calaverite, hessite and altaite, which exist in ore fissures and gangue minerals and are encapsulated in metal sulfide minerals; the size of the Au, Ag, and Te minerals was small the particle size of the natural gold, calaverite and hessite are below 0.038 mm. The ore type was primary gold ore containing polymetallic sulfide with a quartz vein.

The comprehensive recovery of telluride-type gold ores is suitable for low alkalinity condition, and ethyl sulfur nitrogen is suitable as a collector for hessite and altaite, both of which are more closely related to galena. Xanthate collectors work well for collecting Au and Ag.

When a primary gold ore containing polymetallic sulfide with quartz vein was processed under the new technology, the average grades of the Te, Au and Ag were 241.61, 90.30, and 92.74 g/t in the flotation concentrate, and the recovery rates were 95.42%, 97.28%, and 94.65% respectively, so excellent technical indicators were obtained. Compared with the original flotation process, the recovery rates of the Te, Au, and Ag increased by 19.91%, 6.93%, and 5.67%. The comprehensive recovery of Te, Au and Ag in telluride-type gold mines can thus be achieved.

**Author Contributions:** W.Y. and G.W. conceived of and designed the experiments; W.Y. prepared the samples and performed the experiments; W.Y., G.W., Q.W., and H.C., analyzed the data; W.Y., G.W., Q.W., K.Z., and P.D. contributed to the writing and revising of the paper.

**Funding:** We acknowledge the financial support provided by the National Natural Science Foundation of China (Grant No. 51474169), the Shaanxi Provincial Institute of Industrial Science and Technology (Grant No. 2016GY-154), the Key Laboratory of Gold and Resources in Shaanxi Province, Xi'an, China (-710055), the Education Department of Shaanxi Province Key Laboratory Project (Grant No. 18JS061) and the Key Science and Technology Program of Shaanxi Province, China (Grant No. 2018GY-080).

**Conflicts of Interest:** The authors declare no conflict of interest.

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
