# Peer review of "Comprehensive Recovery Technology for Te, Au, and Ag from a Telluride-Type Refractory Gold Mine"

_minerals, doi:10.3390/min9100597_

Round 1
Reviewer 1 Report
Dear editors:
The authors have made the great effort on improve the quality of draft. The revised manuscript has provided the significant improvement on presentation and analysis of data with reasonable discussion. Overall is an interesting work but still need extra effort on the proof-reading for draft.
Author Response
Reviewer 1
The authors have made the great effort on improve the quality of draft. The revised manuscript has provided the significant improvement on presentation and analysis of data with reasonable discussion. Overall is an interesting work but still need extra effort on the proof-reading for draft.
Response: We are very grateful for your positive comments on our manuscript and thank you very much for your careful review and constructive suggestions. Meanwhile, we have invited professional English editors to revise our manuscript carefully.
Reviewer 2 Report
Dear authors,
The flotation separation you have shown in your manuscript is innovative. Certainly a contribution to industrial production, as you have shown. However, you have not given any expression to the flotation kinetics. You have experiments in measuring or quantifying the concentration of minerals, but you do not have time intervals for certain periods. Further, I consider that a similar, if not identical, result of effective selectivity would be achieved with some other thion-carbamate. For example, O-ethyl-N-propyl or O-propyl-N-propyl or ethyl thioncarbamate. Probably, it should be done with some other alkyl xanthate, for example N-hexyl. Finally, I think we need to give a cumulative curve to the granulation of the ground material and do some more testing on the choice of granulation
Author Response
Reviewer 2
The flotation separation you have shown in your manuscript is innovative. Certainly a contribution to industrial production, as you have shown. However, you have not given any expression to the flotation kinetics. You have experiments in measuring or quantifying the concentration of minerals, but you do not have time intervals for certain periods.
Response: Thank you for your suggestion. In the process of mineral flotation, appropriate pulp concentration is beneficial to increase the flotation rate. In this paper, the influence of slurry concentration on the recovery of valuable minerals is mainly considered. However, we mainly studied the optimization of flotation reagent system and flotation process, so we did not show the results of flotation kinetics in the paper. Data and conclusions on flotation kinetics will be presented in future work and papers.
Reviewer 3 Report
The following points need to be revised.
1. Section 3.2.2. The effect of solution pH on the recovery of the elements should be explained.
2. Some comments should be added on the recovery process of pure Au, Ag and Te metal from the concentrates.
Author Response
Reviewer 3
Comment 1: Section 3.2.2. The effect of solution pH on the recovery of the elements should be explained.
Response: Thank you for your suggestion. According to suggestion, we have already explained the effect of solution pH on the recovery of the elements in the revised manuscript. The revised contents are as follows: “pH value has the greatest effect on the recovery of Au, but has little effect on the recovery of Ag”.
Comment 2: Some comments should be added on the recovery process of pure Au, Ag and Te metal from the concentrates.
Response: Thank you for your suggestion. At present, there are two main ways to recover valuable minerals from the telluride-type gold concentrate mine. One is pyrometallurgical, which is to recover Te from flue gas and Au and Ag from roasting slag in the process of roasting concentrate. The other is hydrometallurgy, where Te is leached by alkaline leaching and Au and Ag by cyanidation. Consider the structure and content of the article, which is not presented in the manuscript.
This manuscript is a resubmission of an earlier submission. The following is a list of the peer review reports and author responses from that submission.
Round 1
Reviewer 1 Report
I very much enjoyed reading this paper. There is a lot of information. However, it does lack full explanation of methods and analysis. There also is a need for more imagery from the MLA. I think you should investigate alternative ways to show your grade recovery against grind size. It will be a useful paper with some additions and changes in my opinion.
Line 21 thus, thus...
2.1 - a brief explanation as to why those specific reagents were used would be useful.
Can you explain why 3mm was chosen as the size fraction, that's very coarse?
2.23 - why did you blend the products?
3.1. You need to specify exactly what these analytical techniques do and on what i.e. XRF defines assay - with what limitations? "The S4 PIONEER with advanced 4 kW excitation technology provides highest sensitivity esp. for light elements and traces due to optimized beam geometry." Bruker Pioneer S4. What SEM was the MLA software applied? Any information on the set up of the MLA i.e. WD/voltage/spot size... It may be important to some readers and also provides insight into the 'trickiness' of the mineral texture, it's associations and potential for liberation.
Line 81 - you know roughtly the particle size as you've scalped a 3mm size fraction, do you mean to comment upon the grain size of the mineral of interest?
Line 94 - how large?
Line 99 - what is "blend"? Sphalerite?
Line 990 - can you proved some BSE and or classified -false colour images of the ore particles with labels?
I suggest if you have no dissemination size, that there is no need to add this information into the table.
Line 102 - Please add in that these images are from optical cross-polar investigations and provide the manufacturer of the microscope.
Consider showing your con grade recovery as columns not series, that would be a standard way to present these data.
Line 207 - just simplify XRD - modal mineralogy. mineralogical assemblages, individual minerals. Is XRD really adding value at this point, it's only accurate to 2wt%?
Line 224, these minerals should be cited in the Abstract.
Author Response
Comments: I very much enjoyed reading this paper. There is a lot of information. However, it does lack full explanation of methods and analysis. There also is a need for more imagery from the MLA. I think you should investigate alternative ways to show your grade recovery against grind size. It will be a useful paper with some additions and changes in my opinion.
Response: We are very grateful for your positive comments on our manuscript. In the revised draft, we have detailed the methods and steps of analysis and testing. Your suggestions are of great importance and significance for our future research. Thanks again!
Comment 1: 2.1 - A brief explanation as to why those specific reagents were used would be useful.
Response: Thanks for your valuable suggestion. We have already explained the use of those specific reagents in the revised manuscript. The specific renovation is as follows: Butyl xanthine and sodium isoamyl xanthate are commonly used as sulfide ore collectors; ammonium dibutyl dithiophosphate can improve the recovery of refractory ores, and ethyl thio carbamate is often used in the flotation of galena. (Section 2.1, line 5)
Comment 2: Can you explain why 3mm was chosen as the size fraction, that's very coarse?
Response: This is our mistake and we have made changes in the revised draft. What we want to say is as follows: In the test, raw samples, after they were crushed to less than 3 mm in a roller crusher, were finely ground to -0.074 mm for 60-80% with a ball mill before flotation.
Comment 3: 2.23 - Why did you blend the products?
Response: We've got it wrong here, and we have deleted the “blend” in the revised manuscript.
Comment 4: 3.1. You need to specify exactly what these analytical techniques do and on what i.e. XRF defines assay - with what limitations? "The S4 PIONEER with advanced 4 kW excitation technology provides highest sensitivity esp. for light elements and traces due to optimized beam geometry." Bruker Pioneer S4. What SEM was the MLA software applied? Any information on the set up of the MLA i.e. WD/voltage/spot size... It may be important to some readers and also provides insight into the 'trickiness' of the mineral texture, it's associations and potential for liberation.
Response: Thank you for your valuable suggestion. We have added this section.
With the help of an OPTON mineral liberation analyzer (MLA) (Beijing Opton Optical Technology Co., Ltd.) and an APOLLO X-ray energy dispersive spectrometer (America), the mineral composition, particle size and ore association relation of the telluride-type gold mine experimental sample were measured, and the MLA operating voltage is 25kV during the test. With the help of an S4 Pioneer wavelength dispersive X-ray Fluorescence (XRF) spectrometer (Germany), the mineral elements and content of the mine experimental sample were measured, and the test sample needs to be in a dry state, the weight is above 3g, and the fineness is -0.074 mm. (Section 2.3)
Comment 5: Line 81 - you know roughtly the particle size as you've scalped a 3mm size fraction, do you mean to comment upon the grain size of the mineral of interest?
Response: Thank you for your question. Your explanation is correct. It is through analyzing the grain size of minerals in the ore that we judge the floatability of such minerals and prevent valuable minerals from being over-milled in the grinding process.
Comment 6: Line 94 -how large?
Response: Thank you for your question, we have already added a supplementary explanation. The particle size of pyrite, galena and chalcopyrite is larger than 0.016 mm, and some of them have reached 1~2 mm. (Section 3.1, paragraph 2, line 10)
Comment 7: Line 99 - what is "blend"? Sphalerite?
Response: Thank you for pointing out the word. We have corrected it to “Sphalerite”. (Table.1)
Comment 8: Line 99 - can you proved some BSE and or classified -false colour images of the ore particles with labels?
Response: Thank you for your valuable suggestion. It would be better for us to adopt your suggestion. The MLA we conducted only provided the test results, but the figure was not provided. We cannot present the picture completely, which will be paid more attention to in the future study and work.
Comment 9: I suggest if you have no dissemination size, that there is no need to add this information into the table.
Answer: Thank you for your valuable suggestion. We have revised the table1. 11 ore minerals have been identified, so it is necessary to show them in the table. (Table.1)
Comment 10: Line 102 - Please add in that these images are from optical cross-polar investigations and provide the manufacturer of the microscope.
Response: Thank you for your valuable suggestion. We have added this section. And the polarizing microscope is made in Shanghai Wanheng Precision Instrument Co, Ltd. and the model number is MM-7C. (Section 2.3)
Comment 11: Consider showing your con grade recovery as columns not series, that would be a standard way to present these data.
Response: Thank you for the comments on the paper. We have revised the Figure 3-8 in the manuscript. (Figure 3-8)
Comment 12: Line 207 - just simplify XRD - modal mineralogy. Mineralogical assemblages, individual minerals. Is XRD really adding value at this point, it's only accurate to 2wt%?
Response: Thank you for your question. We have recalculated this data and revised the manuscript. It is shown in Figure 9 that pyrite is the main mineral in flotation concentrate, with a high content of 84%. It also contains a small amount of galena and pyrrhotite, but gangue mineral content is less. (Section 3.4, paragraph 2, line 2)
Comment 13: Line 224, these minerals should be cited in the Abstract.
Response: Thank you for your valuable suggestion. It's already cited in the Abstract in the revised manuscript. (Abstract, line 6)
The reviewers are very kindhearted. We deeply appreciate your helpful suggestions to improve the quality of our manuscript. Thank you for the many related papers you have recommended to us. These papers have been of great help to us. We are looking forward to your positive affirmation on our revised manuscript. Thank you very much.

Reviewer 2 Report
Dear Editorials:
This manuscript by Yang et al. conducted batch and industrial scale flotation tests on comprehensive recovery of Te, Au and Ag from telluride-type gold deposits. The recovery rate of all three major metals are encouraging. However, the authors need provide further data and analysis to support their claims how the better flotation performance achieved regarding with the mineralogy, phase compositions of both concentrates and tailings. Specifically, I have the following comments:
1) The first thing is about the repeatability of the tests, all data and graphs presented in the draft did not include any error range with significance level. Especially on the recovery rate of data, such as table 3, the Ag recovery rate is 94.64%, 94.57%, 95.27%and 94.98% for four different collectors. Four data set has only 0.6% difference. In addition, in section 2.2, the authors did not provide details how to determine the concentration and calculated the rate.
2) In Table 1, Silver should be hessite.; How to determine the particle sizes in table 1?;
3) Please provide the scale bar for all the figures in Figure 2. Based on Figure 2, I would consider the natural gold is strongly associated with quartz, not pyrite. Please provide the reason.
4) In experimental section, please add the extra section on ore and samples characterisation which should include on which facilities and how to conduct ICP, XRF, MLA, Electron Microscopy and XRD analysis.
5) In section 3.3, why the base case chose the condition of pH=9 with 66% of -74 micron particle sizes instead of pH=8 and with 70% of -74 micron . As in the Figure 3 and 4, the data already indicate the pH=9 and 66% are the non-optimal conditions. So the reason for the new process has the better performance may not be due to using new combined collectors but the pH values or particle sizes. So the in the conclusion part, the claim of “The combined collector (sodium isoamyl xanthate + ethyl thio carbamate) had a better recovery of Te, Au and Ag than that in the single reagents.” is plausible.
6) In section 3.4, the authors mentioned that using the XRD determine the phase composition of tailings and concentrate. If the authors can provide the more tailings and concentre of XRD samples, specially for the section 3.2.3, role of ethyl thio carbamate on the recovery of galena can be clearly demonstrated using quantitative XRD for six concentrates.
Even being through English language editing by MDPI, there are still some writing, editing and formatting problems need to be refined. Overall is an interesting work but need authors put more effort on their manuscript.
Author Response
Comments: This manuscript by Yang et al. conducted batch and industrial scale flotation tests on comprehensive recovery of Te, Au and Ag from telluride-type gold deposits. The recovery rates of all three major metals are encouraging. However, the authors need provide further data and analysis to support their claims how the better flotation performance achieved regarding with the mineralogy, phase compositions of both concentrates and tailings.
Response: Thanks for your positive comments on our manuscript. And we have provided the further data and analysis as you suggested.
Comment 1: The first thing is about the repeatability of the tests, all data and graphs presented in the draft did not include any error range with significance level. Especially on the recovery rate of data, such as table 3, the Ag recovery rate is 94.64%, 94.57%, 95.27% and 94.98% for four different collectors. Four data set has only 0.6% difference. In addition, in section 2.2, the authors did not provide details how to determine the concentration and calculated the rate.
Response: Thank you for your questions. For this experiment, the results showed that the types of agents had little effect on the recovery of silver, but had some effect on the grade and yield of silver, and had obvious effect on the flotation of gold and tellurium.
The flotation concentration is the mass concentration, which has been expressed in this paper. The recovery rate amount was then calculated by using:
Ɛ=βγ/β1
Where Ɛ is the element of recovery (%); β is the grade of the element in the product (g/t); γ is the yield of the product (%) and β1 is the grade of feed minerals (g/t). (Section 2.2, line 10)
Comment 2: In Table 1, Silver should be hessite. How to determine the particle sizes in table 1?
Response: Thank you for your question. MLA can detect the mineral content of the ore sample, then mark it on the basis of different color blocks, and then calculate the diameter of the mineral according to the surface area of these different color blocks.
Comment 3: Please provide the scale bar for all the figures in Figure 2. Based on Figure 2, I would consider the natural gold is strongly associated with quartz, not pyrite. Please provide the reason.
Response: Thank you for your valuable suggestion. We have added the scale in figure 2, and we have added Table 3, and Table 3 is known by MLA detection analysis. Among the Table, 48.11% of natural gold and pyrite, as well as 30.40% of natural gold and quartz are concatenated, so natural gold and pyrite are more closely related. (Table 3)
Table 3. Ore Association relation (%)
Ore Association | Natural gold | Calaverite | Silver | Altaite | Antamokite |
Natural gold | - | - | - | - | 0.03 |
Calaverite | - | - | - | 0.69 | 1.7 |
Silver | - | - | - | 1.26 | 11.89 |
Altaite | - | 12.63 | 4.27 | - | 0.56 |
Antamokite | 0.71 | 14.17 | 1.87 | 5.85 | - |
Pyrite | 48.11 | 6.57 | 4.24 | 4.93 | 4.15 |
Galena | - | - | 64.02 | 33.21 | 10.23 |
Chalcopyrite | 2.14 | 1.14 | 0.61 | 5.49 | 0.37 |
Bornite | - | - | - | 0.17 | - |
Quartz | 30.40 | 3.57 | - | 1.08 | 8.03 |
Monomer separation | 18.64 | 61.92 | 24.99 | 47.32 | 63.04 |
Total | 100.00 | 100.00 | 100.00 | 100.00 | 100.00 |
Comment 4: In experimental section, please add the extra section on ore and samples characterisation which should include on which facilities and how to conduct ICP, XRF, MLA, Electron Microscopy and XRD analysis.
Response: Thank you for your valuable suggestion. We have revised the manuscript. With the help of an OPTON mineral liberation analyzer (MLA) (Beijing Opton Optical Technology Co., Ltd.) and an APOLLO X-ray energy dispersive spectrometer (America), the mineral composition, particle size and ore association relation of the telluride-type gold mine experimental sample were measured, and the MLA operating voltage is 25kV during the test. With the help of an S4 Pioneer wavelength dispersive X-ray Fluorescence (XRF) spectrometer(Germany), the mineral elements and content of the mine experimental sample were measured, and the test sample needs to be in a dry state, the weight is above 3g, and the fineness is -0.074 mm. With the help of a MM-7C polarizing microscope (Shanghai Wanheng Precision Instrument Co, Ltd.), the ore association relation of the mine experimental sample was measured. With the help of a PerkinElmer NexION 300X inductively coupled plasma mass spectrometry (ICP-MS) (America), the content of elements of the mine experimental sample were measured, and the test process RF power is 1300 W and the auxiliary gas is 0.7 L/min. With the help of a Ultima IV X-ray diffraction (XRD)(Japan), the mineral composition and content of the mine experimental sample were measured when the test sample weight is above 3 g, fineness is -0.074 mm, the test process is set to 5-90° and the scanning rate is 5°/min. ( Section 2.3)
Comment 5: In section 3.3, why the base case chose the condition of pH=9 with 66% of -74 micron particle sizes instead of pH=8 and with 70% of -74 micron. As in the Figure 3 and 4, the data already indicate the pH=9 and 66% are the non-optimal conditions. So the reason for the new process has the better performance may not be due to using new combined collectors but the pH values or particle sizes. So the in the conclusion part, the claim of “The combined collector (sodium isoamyl xanthate + ethyl thio carbamate) had a better recovery of Te, Au and Ag than that in the single reagents.” is plausible.1
Response: Thank you for your questions. The conditions shown in Figure 8a is the actual process operating conditions and production indicators obtained at the production site of a concentrator in Xiaoqinling before the study. Figure 8b is the industrial test results after the optimization of technological conditions in the study. This is the result obtained by considering the interaction of grinding fineness, pulp pH, collector types, collector dosage and slurry concentration.
Comment 6: In section 3.4, the authors mentioned that using the XRD determine the phase composition of tailings and concentrate. If the authors can provide the more tailings and concentre of XRD samples, specially for the section 3.2.3, role of ethyl thio carbamate on the recovery of galena can be clearly demonstrated using quantitative XRD for six concentrates.
Response: Thank you for your suggestion. For this experiment, under reasonable flotation conditions, the recovery rates of Te, Au and Ag have been greatly improved. but there is no further study that has been conducted on the specific reasons for the effect of the types of collector on Te, Au and Ag recovery. More attention will be paid to this in the future study and work to further investigate the reasons.
Comment 7: Even being through English language editing by MDPI, there are still some writing, editing and formatting problems need to be refined. Overall is an interesting work but need authors put more effort on their manuscript.
Response: Thank you for the comments on the paper. We have revised the manuscript as suggested and have carefully revised the language issues.
The reviewers are very kindhearted. We deeply appreciate your helpful suggestions to improve the quality of our manuscript. Thank you for the many related papers you have recommended to us. These papers have been of great help to us. We are looking forward to your positive affirmation on our revised manuscript. Thank you very much.

Round 2
Reviewer 1 Report
Changes are fine. A nice concise paper.
Reviewer 2 Report
Comment 1: The first thing is about the repeatability of the tests, all data and graphs presented in the draft did not include any error range with significance level. Especially on the recovery rate of data, such as table 3, the Ag recovery rate is 94.64%, 94.57%, 95.27% and 94.98% for four different collectors. Four data set has only 0.6% difference.
The authors failed to reply this main part on the repeatability of the tests and significance level
Comment 2: In Table 1, Silver should be hessite. How to determine the particle sizes in table 1?
The authors failed to correct the Table 1 content "Silver should be hessite"
Comment 3: Please provide the scale bar for all the figures in Figure 2. Based on Figure 2, I would consider the natural gold is strongly associated with quartz, not pyrite. Please provide the reason.
The authors failed to provide the picture to prove the statement" natural gold with pyrite," ,not just by number and statement.
Please provide the scale bar not magnitude number for all the figures in Figure 2 to show the size range of your sample.
Comment 4: In experimental section, please add the extra section on ore and samples characterisation which should include on which facilities and how to conduct ICP, XRF, MLA, Electron Microscopy and XRD analysis.
"With
the help of a Ultima IV X-ray diffraction (XRD)(Japan), the mineral
composition and content of the mine experimental sample were measured
when the test sample weight is above 3 g, fineness is -0.074 mm, the
test process is set to 5-90° and the scanning rate is 5°/min. "
It is plausible that 5 degree/min scan rate can gain the Figure 9 &10 graphs without raw XRD data processing or smoothing given strong content of Fe component (copper source or cobalt source for XRD facility ?). Normal XRD usually only provide the phase information rather than the composition except for quantitative XRD analysis.
Comment 5: In section 3.3, why the base case chose the condition of pH=9 with 66% of -74 micron particle sizes instead of pH=8 and with 70% of -74 micron. As in the Figure 3 and 4, the data already indicate the pH=9 and 66% are the non-optimal conditions. So the reason for the new process has the better performance may not be due to using new combined collectors but the pH values or particle sizes. So the in the conclusion part, the claim of “The combined collector (sodium isoamyl xanthate + ethyl thio carbamate) had a better recovery of Te, Au and Ag than that in the single reagents.” is plausible.1
The conditions shown in Figure 8a is the actual process operating
conditions and production indicators obtained at the production site of
a concentrator in Xiaoqinling before the study. Figure 8b is the
industrial test results after the optimization of technological
conditions in the study.
Even if it is previous industry conditions without optimisation, my comment is that how do you confirm the Figure 8b's optimal result is only due to using new combined collectors. You need provide the evidence for this conlcusion by including other test conditions.